# Recurrent *Campylobacter jejuni* Infections with *In Vivo* Selection of Resistance to Macrolides and Carbapenems: Molecular Characterization of Resistance Determinants

Alexandra Nunes,[a,b] Mónica Oleastro,[a] Frederico Alves,[a*] Nadia Liassine,[c] David M. Lowe,[d] Lucie Benejat,[e] Astrid Ducounau,[e] Quentin Jehanne,[e] Vítor Borges,[a] João Paulo Gomes,[a,b] Gauri Godbole,[f] Lehours Philippe[e,g]

[a]Infectious Diseases Department, National Institute of Health Dr. Ricardo Jorge (INSA), Lisbon, Portugal
[b]Faculty of Veterinary Medicine, Lusófona University, Lisbon, Portugal
[c]Laboratoire Dianalabs, Geneva, Switzerland
[d]University College London, London, United Kingdom
[e]French National Reference Centre for Campylobacters and Helicobacters, Bordeaux Hospital University Centre, Bordeaux, France
[f]UK Health Security Agency, London, United Kingdom
[g]University of Bordeaux, INSERM, Bordeaux Institute of Oncology, Bordeaux, France

Alexandra Nunes and Mónica Oleastro contributed equally to this article. Author order was determined according to authors' contributions to study design, experimental work, management of clinical cases, writing and revision of the manuscript.

**ABSTRACT** We present two independent cases of recurrent multidrug-resistant *Campylobacter jejuni* infection in immunocompromised hosts and the clinical challenges encountered due to the development of high-level carbapenem resistance. The mechanisms associated with this unusual resistance for *Campylobacters* were characterized. Initial macrolide and carbapenem-susceptible strains acquired resistance to erythromycin (MIC > 256mg/L), ertapenem (MIC > 32mg/L), and meropenem (MIC > 32mg/L) during treatment. Carbapenem-resistant isolates developed an in-frame insertion resulting in an extra Asp residue in the major outer membrane protein PorA, within the extracellular loop L3 that connects $\beta$-strands 5 and 6 and forms a constriction zone involved in $Ca^{2+}$ binding. The isolates presenting the highest MIC to ertapenem exhibited an extra non-synonymous mutation (G167A|Gly56Asp) at PorA's extracellular loop L1.

**IMPORTANCE** Carbapenem susceptibility patterns suggest drug impermeability, related to either insertion and/or single nucleotide polymorphism (SNP) within *por*A. Similar molecular events occurring in two independent cases support the association of these mechanisms with carbapenem resistance in *Campylobacter* spp.

**KEYWORDS** *Campylobacter jejuni* recurrent infection, immunocompromised patients, macrolide resistance, high-level carbapenem resistance, major porin

Campylobacteriosis is the most common bacterial gastrointestinal infection worldwide and the most frequently reported foodborne zoonoses in the European Union (EU) (1, 2). *Campylobacter jejuni* is, by far, the species most frequently associated with human infections, being found in most domesticated animal species involved in the human food chain (2, 3).

Symptomatic infections in immunocompetent individuals are mainly characterized by self-limiting gastrointestinal signs, usually acute diarrhea that can progress to haematochezia or melena, fever, cachexia, and abdominal pain. Gastrointestinal and systemic sequelae can develop, including Guillain-Barré syndrome, reactive arthritis, and bacteremia (4, 5). Complicated infections can lead to hospitalization and even death (6).

Immunocompromised patients, like those with hypogammaglobulinemia (7), are more prone to develop clinical complications following a *C. jejuni* infection and can

Address correspondence to Mónica Oleastro, monica.oleastro@insa.min-saude.pt.

*Present address: Frederico Alves, Chief Scientific Office, European Food Safety Authority (EFSA), Parma, Italy.

The authors declare no conflict of interest.

acquire a chronic carriage status or develop recurrent infections with symptomatic episodes over several years (8, 9). Most *Campylobacter* infections do not require antimicrobial treatment, but it is usually deemed necessary in patients with severe and prolonged symptomatology or when infections are extraintestinal (6). Given the predisposition of immunocompromised individuals to develop serious clinical presentations, antibiotics are usually administered in such cases (4, 10).

Macrolides and fluoroquinolones are both considered first-line treatment options for campylobacteriosis (11). However, the exposure of *Campylobacter* to these antimicrobials both in human medicine and in the animal production industry has resulted in increasing rates of antimicrobial resistance, especially to fluoroquinolones, in both humans and animals, with resistance rates above 50% (12–15). Macrolides, namely, azithromycin, have recently become more popular as the first line of treatment in gastrointestinal infections, but macrolide resistance has been rising in some countries since the adoption of this practice. Nevertheless, *C. jejuni* resistance rates are still low, usually falling below 10% (11, 16), even if transmissible resistance mechanisms such as erythromycin resistance methyltranferases (*ermB* and *ermN*) have emerged in Asia or in Europe (17).

Multidrug-resistant (MDR) strains have recently emerged in many countries and, while still representing less than a quarter of human infections, may compromise the success of future antibiotic therapies (18). Carbapenems are regarded as a useful antimicrobial class for complicated campylobacteriosis that does not respond to initial treatment (18). Contrary to other $\beta$-lactams, carbapenem resistance is yet to be defined, with few studies suggesting the possibility of resistance acquisition as a consequence of prolonged selective pressure (19).

The present study aimed to investigate the molecular mechanisms associated with *in vivo* development of antibiotic resistance in two cases of *C. jejuni* infection in immunocompromised hosts, and to assess how the selective pressure can contribute to the development of MDR *Campylobacter* strains, contributing to treatment failure and associated therapeutic challenges.

## RESULTS

**Clinical cases presentation.** We report the investigation of two clinical cases with recurrent multidrug-resistant *C. jejuni* infection in immunocompromised hosts. Case A was a forty-year-old male with Q1-X-linked agammaglobulinemia who developed cellulitis. Two months into treatment with azithromycin for a macrolide-susceptible *Campylobacter* strain from stool, a macrolide-resistant strain was isolated from a blood sample. Switching to ertapenem resulted in a relapse with a carbapenem-resistant strain. The patient was finally successfully treated for *C. jejuni* infection with selective intestinal decontamination with oral neomycin. Seven isolates (designated A1 to A7) were obtained between August 2009 and June 2010, four of them were from blood and three were from stool samples (Table 1, Fig. 1). In clinical case B, a forty-year-old male with common variable immunodeficiency was empirically treated for diarrhea with cefixime and azithromycin; subsequently, *Campylobacter* spp. was detected in stool sample by PCR. Nine months later, he developed worsening diarrhea; *C. jejuni* was isolated from stool and blood samples. He was initially retreated with azithromycin but this was changed to ertapenem after demonstration of macrolide resistance. Eradication failed despite treatment with over 7 weeks of carbapenems (30 days of ertapenem in two separate courses, 21 days of meropenem); meropenem resistance was noted in later isolates obtained after these antibiotic regimens. He was eventually successfully treated with a combination of prolonged tigecycline, chloramphenicol and gentamicin. Five isolates (designated B1 to B5) were obtained between September and December 2016, three of them were from blood and two were from stool samples (Table 1, Fig. 1).

**Resistance phenotypes.** All *C. jejuni* isolates from both cases were resistant to erythromycin (with exception of the susceptible isolate A1), ciprofloxacin, ampicillin, and tetracycline (Table 1). However, following the azithromycin-empirical treatment, the second isolate from case A, isolate A2 from blood sample, became highly resistant to erythromycin (MIC $\geq$256 mg/L). Contrary to B1 and B2 isolates that harbored a low level of resistance to

**TABLE 1** Clinical and phenotypical data of the *Campylobacter jejuni* studied isolates[a]

| | *C. jejuni* isolates_Id | MLST_ST (clonal complex) | Biological sample (isolation source) | Isolation date | Antimicrobial susceptibility (MICs in mg/L) | | | | | | |
|---|---|---|---|---|---|---|---|---|---|---|---|
| | | | | | Ery | Cip | Tet | Amp | Imp | Ert | Mer |
| Case 1 | A1 | 1709 (ST-1034 | Blood | 26-08-2009 | 0.19 | ≥32 | ≥256 | 24 | 0.032 | 0.047 | 0.023 |
| (40-yr-old-male) | A2 | clonal complex) | Blood | 11-11-2009 | ≥256 | ≥32 | ≥256 | 24 | 0.032 | 0.094 | 0.023 |
| | A3 | | Stool | 13-11-2009 | ≥256 | ≥32 | 96 | 24 | 0.032 | 0.064 | 0.016 |
| | A4 | | Blood | 12-01-2010 | ≥256 | ≥32 | ≥256 | 32 | 0.064 | 64 | 8 |
| | A5 | | Blood | 18-01-2010 | ≥256 | ≥32 | ≥256 | ≥256 | 0.19 | 128 | 32 |
| | A6 | | Stool | 05-03-2010 | ≥256 | ≥32 | ≥256 | ≥256 | 0.094 | 64 | 8 |
| | A7 | | Stool | 15-06-2010 | ≥256 | ≥32 | ≥256 | 32 | 0.032 | 0.38 | 0.047 |
| Case 2 | B1 | 1233 (ST-353 | Blood | 12-09-2016 | ≥256 | ≥32 | 32 | ≥256 | 0,25 | 2 | 0,25 |
| (40-yr-old-male) | B2 | clonal complex) | Blood | 02-11-2016 | ≥256 | ≥32 | ≥256 | ≥256 | 0,125 | 2 | 0,25 |
| | B3 | | Stool | 18-11-2016 | ≥256 | ≥32 | ≥256 | ≥256 | 0,125 | ≥512 | 8 |
| | B4 | | Blood | 30-11-2016 | ≥256 | ≥32 | ≥256 | ≥256 | 0,125 | ≥512 | 8 |
| | B5 | | Stool | 14-12-2016 | ≥256 | ≥32 | ≥256 | ≥256 | 0,25 | ≥512 | 32 |

[a]Cutoff values used for strain categorization as resistant (R) were 4 mg/L for erythromycin (Ery), 0.5 mg/L for ciprofloxacin (Cip), 2 mg/L for tetracycline (Tet), 16 mg/L for ampicillin (Amp), and 1 mg/L for ertapenem (Ert) (https://www.sfm-microbiologie.org/wp-content/uploads/2022/05/CASFM2022_V1.0.pdf), and as susceptible (S) <2 mg/L or R > 8 mg/L for meropenem (Mer) and S < 2 mg/L or R > 4 mg/L for imipenem (Imp) (https://www.eucast.org/fileadmin/src/media/PDFs/EUCAST_files/Breakpoint_tables/v_13.0_Breakpoint_Tables.pdf).

ertapenem (MIC = 2 mg/L), case A isolates (A1 to A3) initially exhibited carbapenem susceptibility. However, during prolonged treatment with carbapenems, escalating resistance to ertapenem and to meropenem developed in both cases (Table 1). Indeed, isolates A4 to A6 as well as isolates B3 to B5 acquired high-level ertapenem resistance with MICs ranging from 64 to 128 mg/L and ≥512 mg/L, respectively, while also developing resistance to meropenem, with MIC values increasing from 8 mg/L (A4/A6 and B3/B4) to 32 mg/L (A5 and B5). None of the isolates developed resistance to imipenem, although all isolates resistant to ertapenem and meropenem from case A had approximately 2- to 6-fold higher imipenem MICs than the susceptible ones.

**Analysis of genetic variability.** *In silico* MLST analysis revealed that all isolates from case A belong to ST-1709 (ST-1034 clonal complex), while isolates from case B belong to ST-1233 (ST-353 clonal complex). Using STRUCTURE software (20) and *C. jejuni* host-segregating loci as previously described by Thepault et al. (21), the entire set of isolates was attributed to the chicken population with attribution scores of 100% (data not shown). Although some phage remnant sequences were found (data not shown), no differential accessory genome was observed within either group of isolates, regarding the first isolate collected from each clinical case.

Regarding microevolutionary analysis (see Supplemental files), several mutations, affecting or not phase variable genes, were observed inter- or intra-isolates of both clinical cases,

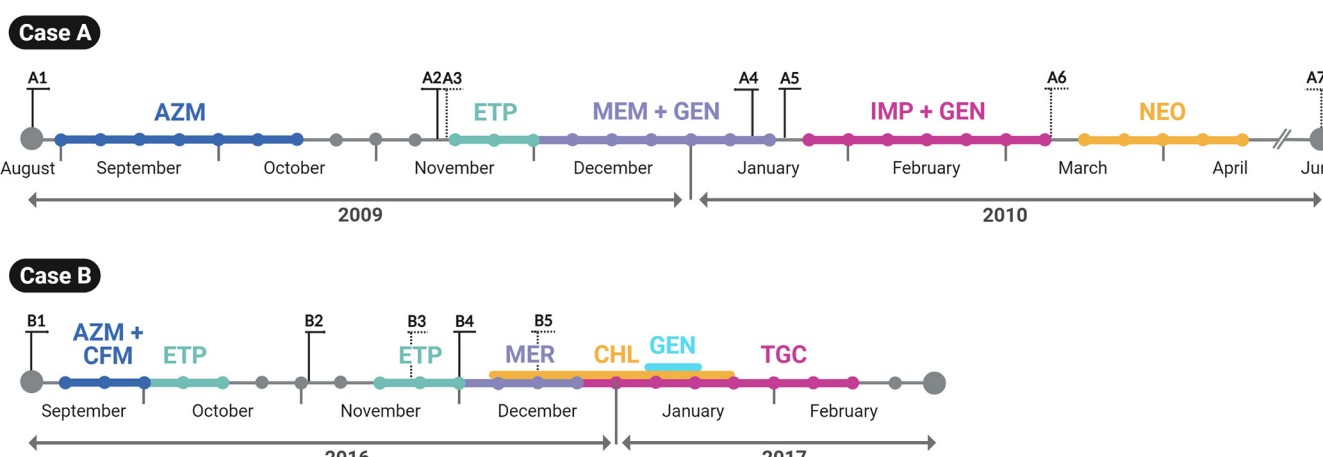

**FIG 1** Treatment timeline of the patients from case A and case B recurrent infections by *Campylobacter jejuni*. For both cases, bacterial strains isolated from blood or fecal samples are indicated in full and dotted lines, respectively, while the antibiotics used and their duration are shown in colored bold horizontal lines (each bullet corresponds to a week). AZM, azithromycin; ETP, ertapenem; MEM, meropenem; GEN, gentamicin; NEO, neomycin; CFM, Cefixime; CHL, chloramphenicol; TGC, tigecycline.

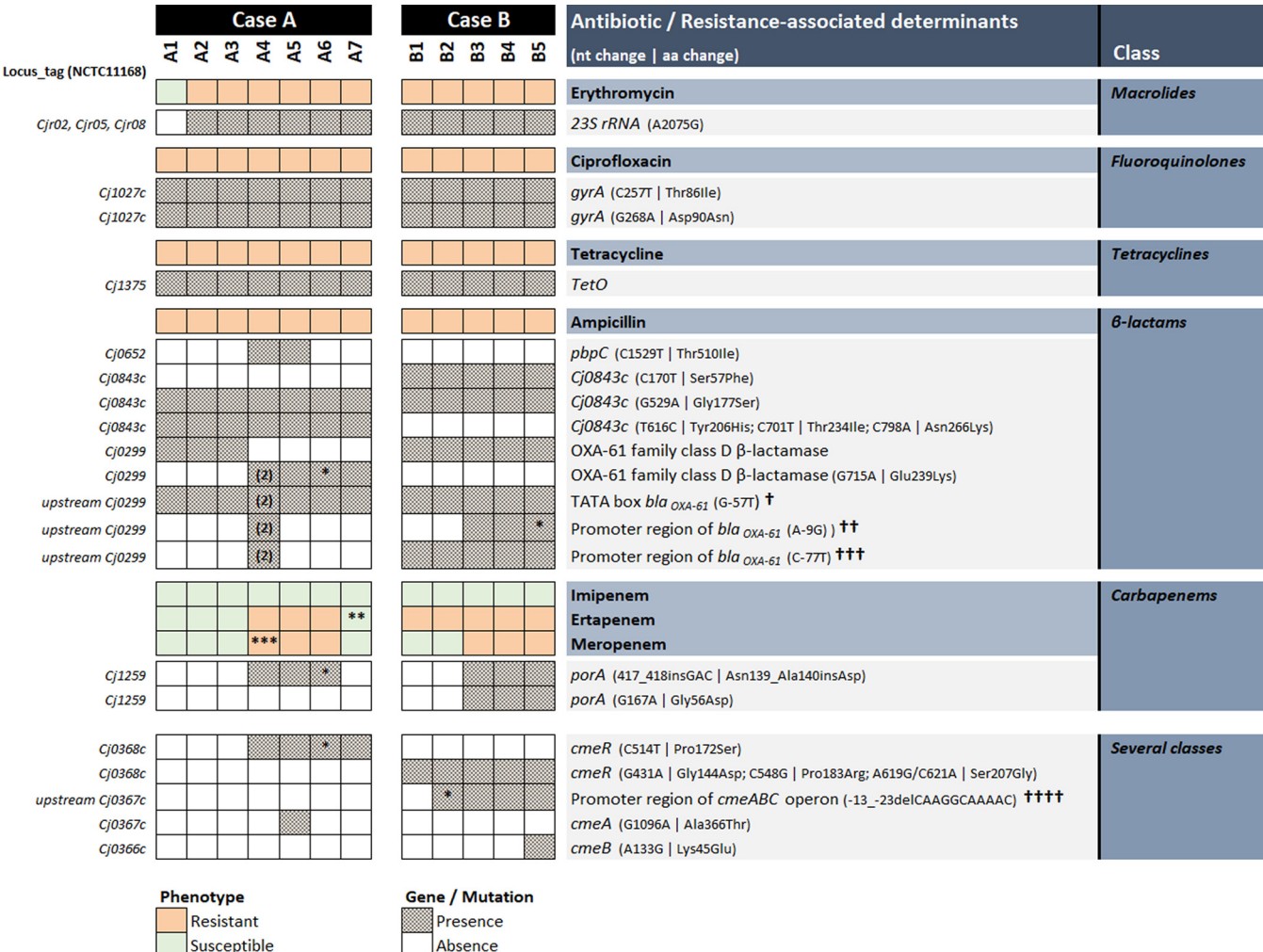

**FIG 2** Known and putative genetic determinants of antibiotic resistance in *Campylobacter jejuni* strains recovered from two cases of recurrent infections. (n), number of copies; *, allele mixture; **, MIC 10× higher than other susceptible strains (see Table 1); ***, heterogeneous resistance; †, restores TATA box from GAAAAT to TAAAAT in the -10 promoter region; ††, restores the *Campylobacter jejuni* ribosome binding site (RBS) consensus sequence (AAGGA); †††, affects the *Campylobacter jejuni* -35 consensus sequence (TTTAAGTnTT); ††††, leads to the loss of a second RBS.

which are likely important for human colonization, infection and adaptability to the immunocompromised host. In agreement with this, all isolates harbored a functional version of the *cipA* gene, which was previously associated with *C. jejuni* persistent human infection (22). A more detailed description of these results can be found in supplemental material and in Tables S3, S4, S5 and Fig. S1.

**Analysis of antibiotic-resistant determinants.** Regarding antibiotic resistance (Fig. 2), known genetic determinants were found in all isolates conferring resistance to erythromycin, ciprofloxacin, and tetracycline, involving point mutations in the three copies of *23S rRNA* (A2075G), double mutations in *gyrA* (C257T|Thr86Ala and G268A|Asp90Asn), and the presence of the *tetO* gene, respectively. Concerning β-lactams, all strains were resistant to ampicillin, although with differing resistance degrees (MICs varying from 24 to ≥256 mg/L), and all carried a G to T transversion upstream of the β-lactamase gene *bla*<sub>oxa-61</sub> that restores the TATA box from GAAAAT to TAAAAT in the -10 region (23, 24). An additional 100% fixed nonsynonymous mutation (G715A|Glu239Lys) was found in *bla*<sub>oxa-61</sub> for A4, A5, and A7 isolates from case A as well as an extra gene copy exclusively for A4, but with no relation with higher MIC values. Isolates from case B, all with ampicillin MIC ≥ 256 mg/L, as well as isolate A4 (MIC = 48 mg/L) presented additional mutations affecting the promoter region of *bla*<sub>oxa-61</sub> (Fig. 2).

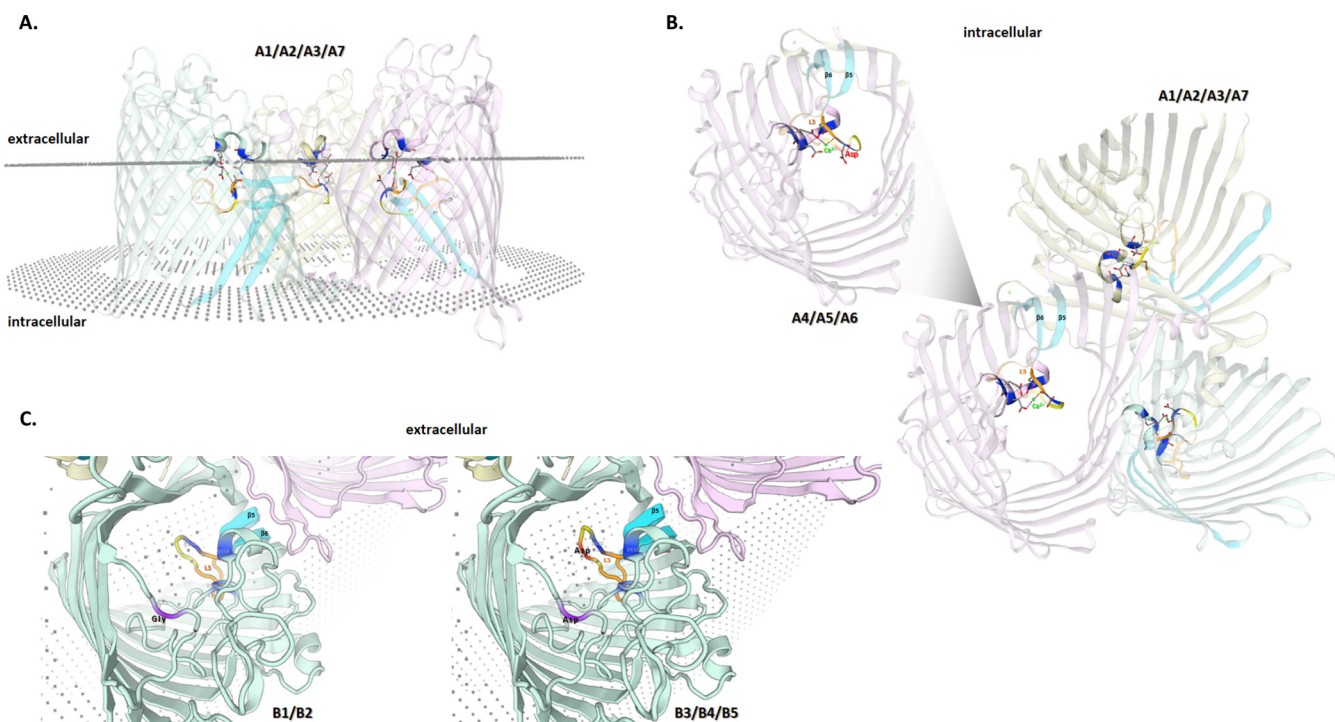

**FIG 3** *In silico* predicted MOMP 3D structure for *Campylobacter jejuni* isolates from clinical case A and case B. All putative trimeric structures were predicted with the SWISS-MODEL server, after signal peptide cleavage (amino acids 1 to 22). For all panels, key features were colored for better visualization of the identified MOMP mutations (see Fig. 2 for details): $\beta$5 and $\beta$6 strands (light blue), extracellular loop L3 (orange), calcium (green sphere), and putative calcium-ligands (blue amino acid residues). For A1/A2/A3/A7 isolates, the trimer is shown (A) viewed from the side regarding the outer membrane, and (B) viewed from the inside the cell (looking out). For A4/A5/A6 isolates, only a representative monomer is displayed with the extra Asp residue depicted in red. For case B isolates (C), representative monomers are displayed viewed from the outside the cell (looking in), with the extra Asp residue depicted in red and the amino acid substitution from Gly to Asp marked in purple.

Apart from isolates A3 and A6 (showing allelic mixture, and therefore excluded from this analysis), all isolates resistant to ertapenem and meropenem exhibited a fixed insertion of a GAC codon between nucleotide positions 417 and 418 of the *porA* gene, coding for *C. jejuni* major outer membrane protein (MOMP), resulting in the inclusion of an extra Asp residue in position 139 of the protein (Fig. 2). Interestingly, no sequence match was found after BLAST nt search, revealing that this in-frame insertion, which developed in both clinical cases after prolonged carbapenem treatment, was never previously reported. According to the *C. jejuni* MOMP structure (25), this insertion falls within the extracellular loop L3 that connects $\beta$-strands 5 and 6 and, together with loops L4 and L6, forms a constriction zone involved in $Ca^{2+}$ ion binding (Fig. 3). Besides this 3-bp insertion, an extra nonsynonymous mutation (G167A|Gly56Asp) at MOMP's extracellular loop L1 was exhibited by the three isolates presenting the highest MIC to ertapenem (isolates B3 to B5, MIC $\geq$ 512 mg/L) (Fig. 2), which, like loops L3, L4 and L6, fold inside the barrel. In general, besides the 3-bp insertion, all carbapenem-resistant isolates possess novel *porA* sequences, differing by 18 and two single nucleotide polymorphisms (SNPs) to their PubMLST closest match allele for clinical case A (allele 1912) and case B (allele 172), respectively. Altogether, it is likely that the observed genetic alterations in *porA* are contributing to resistance to both ertapenem and meropenem.

In addition, unique mutations occurring in the *cmeABC* operon (Fig. 2), which codes for the *C. jejuni* resistance-nodulation-cell division (RND) efflux system, were also observed in the two isolates presenting the highest-level resistance to carbapenems: A5 (MIC for ertapenem = 128 mg/L, MIC for meropenem = 32 mg/L) and B5 (MIC for ertapenem $\geq$512 mg/L, MIC for meropenem = 32 mg/L). Indeed, while the A5 isolate displayed a nonsynonymous mutation (G1096A|Ala366Thr) in *cmeA* (coding for the periplasmic fusion protein), B5 had a nonsynonymous mutation (A133G|Lys45Glu) in *cmeB* (coding for an inner membrane efflux transporter). Moreover, an 11-bp deletion

on the *cmeABC* promoter region leading to the loss of a ribosome-binding site (AAGGCA) (26, 27) was found completely established for B3, B4, and B5 isolates, which may affect the expression of the CmeABC efflux pump. Furthermore, three nonsynonymous mutations were additionally observed in the *cmeR* sequence (allele 33) of all case B isolates, which may also contribute to a low-level resistance to ertapenem, but so far, no association with a resistance phenotype was defined. For case A, a nonsynonymous mutation (C514T|Pro172Ser) occurring in *cmeR* was also observed in isolates A4, A5, and A7, but its association with carbapenem resistance is unlikely since isolate A7, obtained from a stool sample, is susceptible to the three carbapenems.

## DISCUSSION

The present study presents the molecular mechanisms associated with the *in vivo* development of resistance to macrolides and carbapenems in two independent cases of recurrent and invasive *C. jejuni* infections in immunocompromised hosts. The inefficient immune response results in an inability to eliminate infecting microorganisms from the gastrointestinal tract, resulting in patients being administered repeated antibiotic therapy (4). Accordingly, both cases were subjected to multiple courses of antibiotics to tackle a lack of response to initial treatments (Fig. 1). Patients were treated with a macrolide upon hospital presentation, and for case B, a combination with a third-generation cephalosporin was preferred. The initial drug choice aligns with the international recommendations to avoid fluoroquinolones as the first therapeutic option in *Campylobacter* infections (11).

Regarding the genetic determinants of resistance to erythromycin, tetracycline, and ciprofloxacin, previously reported mutations have been found in all phenotypically resistant isolates included herein (Fig. 2). The macrolide-resistant isolates harbored the most commonly reported point mutation, at position 2075 of the peptidyl transferase loop in domain V of the three copies of the 23S rRNA, associated with high-level resistance to erythromycin (28). Prolonged exposure to macrolide antibiotics has been proven to influence the acquisition of resistance by resistant clone selection. This adaptive response is better characterized in the animal production sector, due to the former practice of using tylosin and other macrolides in large scale as a prophylactic measure in animal husbandry (29, 30). Even though the use of subtherapeutic dosages appears to have a bigger impact on the selection of resistant clones than therapeutic usage in clinical settings, the prolonged administration of azithromycin in case A may have contributed to the acquisition of high-level erythromycin resistance. Therefore, acquisition of macrolide resistance should be considered during prolonged treatments and highlights the need for monitoring antimicrobial susceptibility after starting treatment. Of note, many immunodeficient patients also receive prophylactic antibiotics, most commonly azithromycin.

All isolates were resistant to ciprofloxacin from the beginning of treatment, suggesting a preacquired resistance at the chicken reservoir level. They all exhibited the Thr86Ala substitution in the gyrase conferring high-level resistance to fluoroquinolones (31), in addition to a second mutation in *gyrA* (G268A|Asp90Asn), which further increased their level of resistance (32).

Similar to fluoroquinolones, tetracycline resistance, associated with the presence of *tetO* (33), was present from the beginning and was likely acquired in the chicken reservoir as well.

Even though the isolates from case A presented different MICs for ampicillin throughout the course of treatment, $\beta$-lactam resistance was observed in all isolates from both patients. Despite it being a $\beta$-lactamase-encoding gene, the presence of the *bla*$_{oxa-61}$ gene alone is not sufficient to confer a resistant phenotype to *C. jejuni*, as strains carrying this gene are susceptible to ampicillin (34). The $\beta$-lactamase-conferred resistance seems to be related to mutations in the upstream region of the *bla*$_{oxa-61}$, like the G to T transversion found in all isolates of this study, which results in an upregulation of the downstream gene, responsible for the high-level $\beta$-lactam resistance observed (23). The additional

mutations in the promoter region of all isolates from case B, and in isolate A4, may have contributed to the higher MIC detected. On the contrary, the additional mutation of the $bla_{oxa-61}$ gene observed in isolates A4, A5, and A7 (G715A|Glu239Lys), as well as the extra gene copy in isolate A4, does not seem to correlate with ampicillin resistance, as noted by the different MICs of these isolates (Table 1).

Efflux pump systems, of which the CmeABC is the predominant one in *Campylobacter*, may work synergistically with other resistance mechanisms by actively removing a given antibiotic from the bacterial cytosol (28, 35, 36). In the present study, all genes from the *cmeABC* operon displayed nonsynonymous mutations, including the transcriptional regulator encoding gene *cmeR*, affecting isolates from both clinical cases, which likely plays a role in the observed antimicrobial resistance to several classes of antibiotics.

Carbapenems are usually reserved for the treatment of serious cases of *Campylobacter* infection, like those complicated by bacteremia (37, 38). The lack of reported resistance to these antibiotics means it is usually the first-line choice in drug-resistant infections (39). The initial isolates considered here (A1, B1) were susceptible to both imipenem and meropenem, with isolate B1 already showing low-level resistance to ertapenem. After the introduction of carbapenems, the strains' phenotype shifted toward higher resistance (Table 1), suggesting an *in vivo* adaptive response from the bacterial community to the antibiotic selective pressure. According to our genetic analysis, all ertapenem- and meropenem-resistant isolates from both clinical cases developed an in-frame insertion that resulted in the addition of an extra Asp residue (negative charged) in position 139 of MOMP (Fig. 2). To our knowledge, such GAC codon insertion was never described before, so its impact on porin's functionality is unknown. Nevertheless, this unique insertion falls within the L3 extracellular loop, which is highly associated with the MOMP luminal constriction zone (25). Based on the *in silico* predicted MOMP 3D structure for all carbapenem-resistant isolates (Fig. 3), it may disturb the electrostatic balance (due to $Ca^{2+}$ ion binding) established in the protein channel and/or narrow its' constriction zone, thus decreasing bacterial permeability to carbapenem molecules on the basis of their charge and/or size. The effect of pore constriction by amino acid insertions into loop 3 of porins was previously demonstrated for carbapenem resistant *Klebsiella pneumonia* (40). Despite *C. jejuni* usually possessing an innate low susceptibility to larger (>360 Da) dipolar ionic antibiotics (like ertapenem) (41–43), previous studies noting the impact different mutations in the L3 loop have on antimicrobial permeability highlight their importance in emerging antimicrobial resistance (44, 45). Interestingly, an additional nonsynonymous mutation (G167A|Gly56Asp) at the extracellular loop L1 of MOMP was found in the isolates presenting the highest MIC to ertapenem (isolates B3 to B5, MIC >512 mg/L) (Table 1). This extra mutation may further destabilize the MOMP structure by increasing its negative charge, which, by additionally decreasing the membrane permeability to external antimicrobial molecules, contributes to this breakthrough resistance. The two isolates with the highest level of resistance to both ertapenem and meropenem (A5 and B5) (Table 1) revealed two additional mutations affecting the *cmeA* and *cmeB* genes, likely also contributing to carbapenem resistance. Therefore, and similarly to the other classes of antimicrobials herein investigated, a carbapenem resistance mechanism involving the CmeABC efflux pump working synergistically with the aforementioned MOMP changes, is likely to result in higher MIC values, as previously described for other Gram-negative pathogens (44).

Overall, these two cases corroborate previous evidence on the importance of accumulation of independent mutations, which individually have low impact on antibiotic susceptibility, resulting in high-level resistance in a stepwise manner. The dramatic increase in ertapenem MIC values from the initial (2 mg/L) to the later isolates (>512 mg/L), following the use of carbapenems in case B, also supports, for the first time, the role low-level resistance might play in the development of high-level resistance in *Campylobacter*.

The fact that the isolates from both cases either acquired or increased their resistance to carbapenems after their introduction as treatments suggests an adaptive response from the bacterial community through the development of the described resistance mechanisms. The role of the novel mutations described herein on the

development of carbapenem resistance in *Campylobacter* needs to be further explored but emphasizes the need for a rational approach to the use of this critically important class of antimicrobials.

## MATERIALS AND METHODS

**Bacterial culture.** Twelve *C. jejuni* isolates were included in the present study (Table 1): seven isolates (A1 to A7), from clinical case A, were sent to the French National Reference Centre for Campylobacters and Helicobacters ([www.cnrch.fr](www.cnrch.fr)) (P. Lehours) by N. Liassine (Dianalabs); five isolates (B1 to B5), from clinical case B, were sent to the Gastrointestinal Bacteria Reference Unit, UK Health Security Agency (G. Godbole). In both cases, patients provided written consent. Upon receipt, the isolates were subcultured under microaerobic atmosphere (6% O2, 5 to 10% CO2, 80 to 90% N2, 5 to 10% H2) in jars using an Anoxomat microprocessor (Mart Microbiology, B.V. Lichtenvoorde, The Netherlands). Species identification was confirmed by MALDI-TOF mass spectrometry (Bruker Daltonics, Bremen, Germany) (46). All strains were conserved at −80°C in homemade *brucella* broth supplemented with 20% glycerol.

**Antimicrobial susceptibility testing.** Antimicrobial susceptibility testing (AST) was performed according to the European Committee for Antimicrobial and Susceptibility Testing recommendations (47) on commercialized Mueller-Hinton agar MHF (bioMérieux), at 37°C under microaerobic atmosphere. MICs were determined for each isolate with Etest strips (bioMérieux). For erythromycin, ciprofloxacin, tetracycline, amoxicillin, gentamicin, and ertapenem the cutoffs of the "Comité de l'antibiogramme de la Société Française de Microbiologie" (CA-SFM) (V.1.0 Mai 2022) were employed (48); for imipenem and meropenem, generalist EUCAST cutoffs based on PK/PD were used (47). Ertapenem and meropenem MICs were verified by agar dilution method. More details are provided in supplemental materials and methods.

**Whole-genome sequencing, genome assembly and annotation.** High-quality DNA samples (details of DNA extraction are provided in supplemental materials and methods) were quantified using QubitTM (ThermoFisher Scientific) and subjected to dual-indexed Nextera XT Illumina library preparation (Illumina). Libraries were subjected to cluster generation and paired-end sequencing (2x150bp or 2x100bp) on MiSeq or HiSeq2500 Illumina equipment (Illumina).

Genomes were *de novo* assembled using the INNUca v4.2.2 pipeline ([https://github.com/B-UMMI/INNUca](https://github.com/B-UMMI/INNUca)), an integrative bioinformatics pipeline that consists of several integrated modules for reads QA/QC, *de novo* assembly and postassembly optimization steps. Briefly, after reads' quality analysis using FastQC v0.11.5 ([http://www.bioinformatics.babraham.ac.uk/projects/fastqc/](http://www.bioinformatics.babraham.ac.uk/projects/fastqc/)) and cleaning with Trimmomatic v0.38 ([http://www.usadellab.org/cms/?page=trimmomatic](http://www.usadellab.org/cms/?page=trimmomatic)) (49), genomes were *de novo* assembled with SPAdes 3.14.0 ([http://bioinf.spbau.ru/spades](http://bioinf.spbau.ru/spades)) (50) with a mean depth of coverage above 160×, and subsequently improved using Pilon v1.23 (51). Draft genome sequences (of ∼1.61 Mbp in length split in no more than 35 contigs for A1 to A7 isolates and ∼1.71 Mbp in length split in no more than 58 contigs for B1 to B5 isolates) were annotated with RAST server v2.0 ([http://rast.nmpdr.org/](http://rast.nmpdr.org/)). Locus tag of the *C. jejuni* subsp. *jejuni* NCTC 11168 = ATCC 700819 strain (GenBank [NC_002163.1](NC_002163.1)) was adopted to designate all gene hits identified in the subsequent analyses, whenever it was possible.

**Strains' genomic characterization and microevolutionary analysis.** *In silico* multi locus sequence type (MLST) prediction was performed using the *mlst* v2.18.1 software ([https://github.com/tseemann/mlst](https://github.com/tseemann/mlst)), and *porA* typing using the PubMLST platform ([http://pubmlst.org/](http://pubmlst.org/), accessed in October 2022). Potential sources of contamination were estimated using STRUCTURE software v2.3.4 (20) for 15 host-segregating genes, as previously described (21).

To maximize the number of sites available for SNP and indel comparison and potentially provide greater discrimination resolution, trimmed reads were mapped against the draft genome of the first isolate collected from each clinical case with a susceptible phenotype for carbapenems (A1 and B1 isolates were used as reference genomes). Variants were called in sites with minimum mapping quality of 60, minimum base quality of 20, and minimum number of reads covering the variant position ≥10. For each clinical case, all inter- and intrastrain SNPs and indels acquired throughout the microevolution of *C. jejuni* during infection were carefully inspected and confirmed using IGV v2.15.2 ([http://software.broadinstitute.org/software/igv/](http://software.broadinstitute.org/software/igv/)) (52).

For each clinical case, assemblies were aligned using the progressive algorithm of MAUVE software version 2.3.1 ([http://darlinglab.org/mauve/mauve.html](http://darlinglab.org/mauve/mauve.html)) to inspect the accessory genome among isolates, and additionally queried for the existence of phages, mobile genetic elements (MGEs) and plasmids. After confirming the absence of accessory genome, a microevolutionary analysis was carried out using Snippy v4.6.0. Please refer to detailed methods in supplemental materials and methods.

***In silico* identification of virulence and antibiotic resistance determinants.** For all isolates, both the ResFinder 4.1 ([https://cge.cbs.dtu.dk/services/ResFinder/](https://cge.cbs.dtu.dk/services/ResFinder/)) and the Comprehensive Antibiotic Resistance Database 3.2.5 (CARD-RGI 6.0.0) ([https://card.mcmaster.ca/analyze/rgi](https://card.mcmaster.ca/analyze/rgi)) web servers were used to identify mutations and/or genes likely associated with acquired antimicrobial resistance (AMR), using the default parameters (accessed in November 2022).

**Major outer membrane protein structure modulation.** For all isolates from both clinical cases, prediction of the secondary structure of the major outer membrane protein (MOMP), encoded by the *porA* gene, was performed using PRED-TMBB (53) with the posterior decoding method and visualized by TMRPres2D (54), as previously described (55). SignalP-6.0 Server ([https://www.cbs.dtu.dk/services/SignalP/](https://www.cbs.dtu.dk/services/SignalP/)) was used to check the presence of signal peptides and their cleavage sites. Homology modeling of the three-dimensional (3D) MOMP structures was performed with the freely available SWISS-MODEL server ([http://swissmodel.expasy.org/](http://swissmodel.expasy.org/)).

**Data availability.** All raw sequence reads used in the present study were deposited in the European Nucleotide Archive (ENA) under the study accession numbers PRJEB42628 and PRJNA505131 (Table S2). The draft genome assemblies of the first isolate collected from each clinical case (A1 and B1), together with the respective annotation, are provided at https://doi.org/10.5281/zenodo.7684724.

## SUPPLEMENTAL MATERIAL

Supplemental material is available online only.
**SUPPLEMENTAL FILE 1**, DOCX file, 0.03 MB.
**SUPPLEMENTAL FILE 2**, TIF file, 2.2 MB.
**SUPPLEMENTAL FILE 3**, XLSX file, 0.01 MB.
**SUPPLEMENTAL FILE 4**, DOCX file, 0.01 MB.
**SUPPLEMENTAL FILE 5**, XLSX file, 0.02 MB.
**SUPPLEMENTAL FILE 6**, XLSX file, 0.01 MB.
**SUPPLEMENTAL FILE 7**, XLSX file, 0.01 MB.

## ACKNOWLEDGMENTS

This work was partially supported by GenomePT (ref. POCI-01-0145-FEDER-022184) from Fundação para a Ciência e Tecnologia, Portugal. We thank Craig Swift, Gastrointestinal Bacteria Reference Unit, UK Health Security Agency for sequencing the strains from case B.

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
