## [Reviewer comments · Microbiology Spectrum]

Microbiology Spectrum

Recurrent *Campylobacter jejuni* infections with in vivo selection of resistance to macrolides and carbapenems: molecular characterisation of resistance determinants

Alexandra Nunes, Mónica Oleastro, Frederico Alves, Nadia Liassine, David Lowe, Lucie Bénéjat, Astrid Ducournau, Quentin Jehanne, Vítor Borges, João Gomes, Gauri Godbole, and Lehours Philippe

Corresponding Author(s): Mónica Oleastro, National Institute of Health

Review Timeline:

Submission Date:	March 10, 2023
Editorial Decision:	April 14, 2023
Revision Received:	April 20, 2023
Editorial Decision:	May 1, 2023
Revision Received:	May 5, 2023
Accepted:	May 24, 2023

Editor: Cheryl Andam

Reviewer(s): The reviewers have opted to remain anonymous.

Transaction Report:

DOI: <https://doi.org/10.1128/spectrum.01070-23>

April 14, 2023

Prof. Mónica Oleastro
National Institute of Health
Lisboa 1649-016
Portugal

Re: Spectrum01070-23 (Recurrent *Campylobacter jejuni* infections with in vivo selection of resistance to macrolides and carbapenems: molecular characterisation of resistance determinants)

Dear Prof. Mónica Oleastro:

Link Not Available

Sincerely,

Cheryl Andam

Journals Department
Reviewer comments:

Reviewer #1 (Comments for the Author):

Nunes et al. describe two sets of *Campylobacter jejuni* isolates, each set from an immuno-compromised patient that had long-term infections that became resistant to antimicrobials. Resistance to fluoroquinolones, tetracycline and ampicillin were as expected based on the genetic markers. However, resistance to carbapenems had a previously undescribed association with changes in the *porA* gene. The association was very strong, but it should still be recognized that that this is circumstantial evidence.

Line 121: Although attribution score data are not shown, there should be a reference here that shows how that score was developed.

Lines 122 -123: "no differential accessory genome" - different from what?

Line 134: Isolate A1 was susceptible to erythromycin, and the rest were resistant. Was there a change in the rRNA sequences? How was it determined that all three copies of the rRNA gene were the same (assembly of short-read sequences would be hard to distinguish between two and three copies)?

How was it decided what was supplementary material?

What I assume to be Figure S1 was difficult to decipher. Tiny print cannot be read without zooming then you can't see much of the total picture. When you say "boxes," do you mean the labels at the top of each chart or the bars that make up the charts. I do not see a distinction between the red, green, and black bar legend - why are multiple colors used. What is a "main in-length variable polyN tract"?

Line 290: Table S1 is list of polyNtracts - what does that have to do with MICs as stated in this line?

Lines 194 - 195: References from 2007 and 2004 probably do not represent current practices on antimicrobial usage for prophylactics in animal husbandry. There have been dramatic shifts in those practices in the last decade, at least in developed countries.

Lines 313 - 314: In what sense is the phrase "accessory genome" being used? The ordinarily understood sense would be relative to the universal population, in which case there is no bacterial genome that doesn't have some accessory genome. The statement could be correct if it is just referring to the 12 isolates in the study. Clarification is needed.

References are incomplete: 12, 13, 14, 15, 16, 18, 19, 21, 24, 25, 26, 29, 30, 31, 32, 34, 38, 43, 44, 49

Reviewer #2 (Comments for the Author):

This study from Nunes et al. is investigating in vivo development of antimicrobial resistance from two independent infection with *Campylobacter jejuni* strain. The authors demonstrated the potential involved mechanisms as well as the complete characterisation of strains. This study is well conducted and well written. The data showed support adequately the conclusions. I have no major comments to address. Is the authors have verified if the observed phenomenon should be reversible after numerous plating of the bacteria on standard media? Also, bacterial name should be italicized, particularly in the introduction section.

Staff Comments:

Preparing Revision Guidelines

Please return the manuscript within 60 days; if you cannot complete the modification within this time period, please contact me. If you do not wish to modify the manuscript and prefer to submit it to another journal, please notify me of your decision immediately so that the manuscript may be formally withdrawn from consideration by Microbiology Spectrum.

If your manuscript is accepted for publication, you will be contacted separately about payment when the proofs are issued;

please follow the instructions in that e-mail. Arrangements for payment must be made before your article is published. For a complete list of **Publication Fees**, including supplemental material costs, please visit our website.

Spectrum01070-23

We would like to thank the reviewers for the comments that have improved the quality of the manuscript. Please find below the point-by-point responses to the concerns highlighted by the reviewers.

Reviewer #1 (Comments for the Author):

Nunes et al. describe two sets of *Campylobacter jejuni* isolates, each set from an immunocompromised patient that had long-term infections that became resistant to antimicrobials. Resistance to fluoroquinolones, tetracycline and ampicillin were as expected based on the genetic markers. However, resistance to carbapenems had a previously undescribed association with changes in the *porA* gene. The association was very strong, but it should still be recognized that that this is circumstantial evidence.

Line 121: Although attribution score data are not shown, there should be a reference here that shows how that score was developed.

R: Yes, we agree with this comment. The references for material (REF 46) and method (REF 45) for that specific analysis is described in materials and methods section.

To be more specific: for molecular source attribution, we used a method and a subset of 15 host-segregating genes that have been identified in a previous study by Thepault *et al.*, (46). Alleles for each isolate were extracted using Blastn command line tool and were assigned to a number. Using a training dataset of 583 isolates from chicken, ruminant and environment reservoirs, STRUCTURE statistical analysis (45) was conducted in order to attribute each alleles number combination to its potential source.

In order to include these 2 references in the results, the text was adjusted as follows:

“Using STRUCTURE software (45) and *C. jejuni* host-segregating loci as previously described by Thepault *et al.*, (46), the entire set of isolates was attributed to the chicken population with attribution scores of 100% (data not shown).”

Lines 122 -123: "no differential accessory genome" - different from what?

Reply: For each clinical case, we found no differences among isolates, regarding the first isolate collected. We clarified in the text:

Although some phage remnant sequences were found (data not shown), no differential accessory genome was observed within either group of isolates, regarding the first isolate collected from each clinical case.

Line 134: Isolate A1 was susceptible to erythromycin, and the rest were resistant. Was there a change in the rRNA sequences? How was it determined that all three copies of the rRNA gene were the same (assembly of short-read sequences would be hard to distinguish between two and three copies)?

Reply: there was a change in the 23S rRNA sequences, with a 100% the replacement of an A to a G in position 2075 in all resistant isolates. In detail, when we mapped the reads of these resistant isolates against isolate A1, we observed a 100% fixed mutation in this position, with an increase depth coverage that corresponds to 3X the coverage of the genome median coverage. In addition, in order to confirm the existence of the same mutation in the 3 copies, we mapped the reads against individual copies of the 23S rRNA (Cjr02, Cjr05, Cjr08) from a reference genome available at Genbank (NCTC11168). This information is detailed in Figure 1.

How was it decided what was supplementary material?

Reply: that is a very pertinent comment. Phase variation is a major mechanism of creating heterogeneity for host adaptation in *C. jejuni*, and the two described cases of long-term Campylobacter infections nicely illustrated the contribution of these phenomena for the *in vivo* evolution of the bacteria. On the other hand, the main relevance of this study is the development of high-level resistance to carbapenems and the new putative mechanisms associated. Therefore, to not lose the focus on the resistance, authors have decided to place the phase variation sections in supplementary material. However, with the reviewer's comment, we feel that some parts of the material and methods, regarding the bacterial culture, genome assembly and SNP-based microevolutionary analysis, should be placed in the manuscript, in order to make it easier for the readers to follow the experiments and analyses performed. Therefore, these modifications were performed.

What I assume to be Figure S1 was difficult to decipher. Tiny print cannot be read without zooming then you can't see much of the total picture. When you say "boxes," do you mean the labels at the top of each chart or the bars that make up the charts. I do not see a distinction between the red, green, and black bar legend - why are multiple colors used. What is a "main in-length variable polyN tract"?

Reply: As stated in the respective legend, boxes are the colored (blue, yellow and black) labels on the top of each chart. The blue (case A) and yellow (case B) boxes represent all phase variable polyN regions with genetic heterogeneity among isolates, while black boxes (from both cases) represent polyN regions where the main in-length variable polyN tract was found to be conserved among isolates.

We understand the need for an explanation of the colours, and we add the following in Figure S1 legend:

For all polyG/C regions, the green bars represent the "ON" expression status, the red bars represent the "OFF" expression status and the black bars refers to non-coding regions. The grey bars refers to polyA/T tracts that fall in non-coding regions.

The "main in-length variable polyN tract" refers to the predominant tract from a polyN region. For instance, in case B, for locus Cj0318, 9G/C is the main tract for all isolates, except for isolate B1, which shows mixture alleles, in which the 9G/C comprises 50% of the tracts observed.

Line 290: Table S1 is list of polyNtracts - what does that have to do with MICs as stated in this line?

Reply: this was a mistake, we removed including Table S1 from this sentence: more details are provided in supplementary materials and methods.

Lines 194 - 195: References from 2007 and 2004 probably do not represent current practices on antimicrobial usage for prophylactics in animal husbandry. There have been dramatic shifts in those practices in the last decade, at least in developed countries.

Reply: we fully agree with the reviewer concerning current practices on the antimicrobial use in the food animal sector. However, our aim was to emphasize the importance of antibiotic exposure to the development of antimicrobial resistance. In the case of *Campylobacter*, the best model to illustrate this adaptative response is the development of macrolide-resistant mutants that involves a multistep process and requires prolonged exposure to the antibiotic. therefore, and in agreement with the reviewer, we rephrased the sentence, stating that the use of antimicrobials is not a current practice, but instead a former practice.

Lines 313 - 314: In what sense is the phrase "accessory genome" being used? The ordinarily understood sense would be relative to the universal population, in which case there is no bacterial genome that doesn't have some accessory genome. The statement could be correct if it is just referring to the 12 isolates in the study. Clarification is needed.

Reply: Yes, we agree with the reviewer, and we have modified the sentence accordingly:

For each clinical case, assemblies were aligned using the progressive algorithm of MAUVE software version 2.3.1 (<http://darlinglab.org/mauve/mauve.html>) to inspect the accessory genome among isolates, and additionally queried for the existence of phages, mobile genetic elements (MGEs) and plasmids.

References are incomplete: 12, 13, 14, 15, 16, 18, 19, 21, 24, 25, 26, 29, 30, 31, 32, 34, 38, 43, 44, 49

Reply: We thank the reviewer for his care in reviewing the bibliographic references.

All the references listed were reviewed in accordance with the journal's citation rules. The updated references were added using the EndNote output style for ASM Journals provided in the website: https://endnote.com/style_download/american-society-for-microbiology-asm-journals-2/

Reviewer #2 (Comments for the Author):

This study from Nunes et al. is investigating in vivo development of antimicrobial resistance from two independent infection with *Campylobacter jejuni* strain. The authors demonstrated the potential involved mechanisms as well as the complete characterisation of strains. This study is well conducted and well written. The data showed support adequately the conclusions. I have no major comments to address. Is the authors have verified if the observed phenomenon should be reversible after numerous plating of the bacteria on standard media? Also, bacterial name should be italicized, particularly in the introduction section.

Reply: we thank the positive comments regarding our work. We did not evaluate the reversibility of the *porA* in-frame insertion.

May 1, 2023

Prof. Mónica Oleastro
National Institute of Health
Lisboa 1649-016
Portugal

Re: Spectrum01070-23R1 (Recurrent *Campylobacter jejuni* infections with in vivo selection of resistance to macrolides and carbapenems: molecular characterisation of resistance determinants)

Dear Prof. Mónica Oleastro:

Editor's note: It seems you have uploaded the marked up version of the manuscript with the changes, but not the clean version with the changes. The manuscript that was uploaded was the original version (version submitted prior to review). Please upload both the marked up version and clean version of the revised manuscript.

Link Not Available

Sincerely,

Cheryl Andam

Journals Department
Reviewer comments:

Staff Comments:

Preparing Revision Guidelines

To submit your modified manuscript, log onto the eJP submission site at <https://spectrum.msubmit.net/cgi-bin/main.plex>. Go to Author Tasks and click the appropriate manuscript title to begin the revision process. The information that you entered when you first submitted the paper will be displayed. Please update the information as necessary. Here are a few examples of required

updates that authors must address:

Please return the manuscript within 60 days; if you cannot complete the modification within this time period, please contact me. If you do not wish to modify the manuscript and prefer to submit it to another journal, please notify me of your decision immediately so that the manuscript may be formally withdrawn from consideration by Microbiology Spectrum.

Spectrum01070-23

We would like to thank the reviewers for the comments that have improved the quality of the manuscript. Please find below the point-by-point responses to the concerns highlighted by the reviewers.

Reviewer #1 (Comments for the Author):

Nunes et al. describe two sets of *Campylobacter jejuni* isolates, each set from an immunocompromised patient that had long-term infections that became resistant to antimicrobials. Resistance to fluoroquinolones, tetracycline and ampicillin were as expected based on the genetic markers. However, resistance to carbapenems had a previously undescribed association with changes in the *porA* gene. The association was very strong, but it should still be recognized that that this is circumstantial evidence.

Line 121: Although attribution score data are not shown, there should be a reference here that shows how that score was developed.

R: Yes, we agree with this comment. The references for material (REF 46) and method (REF 45) for that specific analysis is described in materials and methods section.

To be more specific: for molecular source attribution, we used a method and a subset of 15 host-segregating genes that have been identified in a previous study by Thepault *et al.*, (46). Alleles for each isolate were extracted using Blastn command line tool and were assigned to a number. Using a training dataset of 583 isolates from chicken, ruminant and environment reservoirs, STRUCTURE statistical analysis (45) was conducted in order to attribute each alleles number combination to its potential source.

In order to include these 2 references in the results, the text was adjusted as follows:

“Using STRUCTURE software (45) and *C. jejuni* host-segregating loci as previously described by Thepault *et al.*, (46), the entire set of isolates was attributed to the chicken population with attribution scores of 100% (data not shown).”

Lines 122 -123: "no differential accessory genome" - different from what?

Reply: For each clinical case, we found no differences among isolates, regarding the first isolate collected. We clarified in the text:

Although some phage remnant sequences were found (data not shown), no differential accessory genome was observed within either group of isolates, regarding the first isolate collected from each clinical case.

Line 134: Isolate A1 was susceptible to erythromycin, and the rest were resistant. Was there a change in the rRNA sequences? How was it determined that all three copies of the rRNA gene were the same (assembly of short-read sequences would be hard to distinguish between two and three copies)?

Reply: there was a change in the 23S rRNA sequences, with a 100% the replacement of an A to a G in position 2075 in all resistant isolates. In detail, when we mapped the reads of these resistant isolates against isolate A1, we observed a 100% fixed mutation in this position, with an increase depth coverage that corresponds to 3X the coverage of the genome median coverage. In addition, in order to confirm the existence of the same mutation in the 3 copies, we mapped the reads against individual copies of the 23S rRNA (Cjr02, Cjr05, Cjr08) from a reference genome available at Genbank (NCTC11168). This information is detailed in Figure 1.

How was it decided what was supplementary material?

Reply: that is a very pertinent comment. Phase variation is a major mechanism of creating heterogeneity for host adaptation in *C. jejuni*, and the two described cases of long-term Campylobacter infections nicely illustrated the contribution of these phenomena for the *in vivo* evolution of the bacteria. On the other hand, the main relevance of this study is the development of high-level resistance to carbapenems and the new putative mechanisms associated. Therefore, to not lose the focus on the resistance, authors have decided to place the phase variation sections in supplementary material. However, with the reviewer's comment, we feel that some parts of the material and methods, regarding the bacterial culture, genome assembly and SNP-based microevolutionary analysis, should be placed in the manuscript, in order to make it easier for the readers to follow the experiments and analyses performed. Therefore, these modifications were performed.

What I assume to be Figure S1 was difficult to decipher. Tiny print cannot be read without zooming then you can't see much of the total picture. When you say "boxes," do you mean the labels at the top of each chart or the bars that make up the charts. I do not see a distinction between the red, green, and black bar legend - why are multiple colors used. What is a "main in-length variable polyN tract"?

Reply: As stated in the respective legend, boxes are the colored (blue, yellow and black) labels on the top of each chart. The blue (case A) and yellow (case B) boxes represent all phase variable polyN regions with genetic heterogeneity among isolates, while black boxes (from both cases) represent polyN regions where the main in-length variable polyN tract was found to be conserved among isolates.

We understand the need for an explanation of the colours, and we add the following in Figure S1 legend:

For all polyG/C regions, the green bars represent the "ON" expression status, the red bars represent the "OFF" expression status and the black bars refers to non-coding regions. The grey bars refers to polyA/T tracts that fall in non-coding regions.

The "main in-length variable polyN tract" refers to the predominant tract from a polyN region. For instance, in case B, for locus Cj0318, 9G/C is the main tract for all isolates, except for isolate B1, which shows mixture alleles, in which the 9G/C comprises 50% of the tracts observed.

Line 290: Table S1 is list of polyNtracts - what does that have to do with MICs as stated in this line?

Reply: this was a mistake, we removed including Table S1 from this sentence: more details are provided in supplementary materials and methods.

Lines 194 - 195: References from 2007 and 2004 probably do not represent current practices on antimicrobial usage for prophylactics in animal husbandry. There have been dramatic shifts in those practices in the last decade, at least in developed countries.

Reply: we fully agree with the reviewer concerning current practices on the antimicrobial use in the food animal sector. However, our aim was to emphasize the importance of antibiotic exposure to the development of antimicrobial resistance. In the case of *Campylobacter*, the best model to illustrate this adaptative response is the development of macrolide-resistant mutants that involves a multistep process and requires prolonged exposure to the antibiotic. therefore, and in agreement with the reviewer, we rephrased the sentence, stating that the use of antimicrobials is not a current practice, but instead a former practice.

Lines 313 - 314: In what sense is the phrase "accessory genome" being used? The ordinarily understood sense would be relative to the universal population, in which case there is no bacterial genome that doesn't have some accessory genome. The statement could be correct if it is just referring to the 12 isolates in the study. Clarification is needed.

Reply: Yes, we agree with the reviewer, and we have modified the sentence accordingly:

For each clinical case, assemblies were aligned using the progressive algorithm of MAUVE software version 2.3.1 (<http://darlinglab.org/mauve/mauve.html>) to inspect the accessory genome among isolates, and additionally queried for the existence of phages, mobile genetic elements (MGEs) and plasmids.

References are incomplete: 12, 13, 14, 15, 16, 18, 19, 21, 24, 25, 26, 29, 30, 31, 32, 34, 38, 43, 44, 49

Reply: We thank the reviewer for his care in reviewing the bibliographic references.

All the references listed were reviewed in accordance with the journal's citation rules. The updated references were added using the EndNote output style for ASM Journals provided in the website: https://endnote.com/style_download/american-society-for-microbiology-asm-journals-2/

Reviewer #2 (Comments for the Author):

This study from Nunes et al. is investigating in vivo development of antimicrobial resistance from two independent infection with *Campylobacter jejuni* strain. The authors demonstrated the potential involved mechanisms as well as the complete characterisation of strains. This study is well conducted and well written. The data showed support adequately the conclusions. I have no major comments to address. Is the authors have verified if the observed phenomenon should be reversible after numerous plating of the bacteria on standard media? Also, bacterial name should be italicized, particularly in the introduction section.

Reply: we thank the positive comments regarding our work. We did not evaluate the reversibility of the *porA* in-frame insertion.

May 24, 2023

Prof. Mónica Oleastro
National Institute of Health
Lisboa 1649-016
Portugal

Re: Spectrum01070-23R2 (Recurrent *Campylobacter jejuni* infections with in vivo selection of resistance to macrolides and carbapenems: molecular characterisation of resistance determinants)

Dear Prof. Mónica Oleastro:

Your manuscript has been accepted, and I am forwarding it to the ASM Journals Department for publication. You will be notified when your proofs are ready to be viewed.

Sincerely,

Cheryl Andam
Editor, Microbiology Spectrum
